# Examining the Usage Patterns of Non-Nutritive Sweeteners among Non-Diabetic Individuals: Insights from the Longitudinal Study of Adult Health (ELSA-Brasil)

**DOI:** 10.3390/nu15224785

**Published:** 2023-11-15

**Authors:** Taiz Karla Brunetti Moreira, Hully Cantão dos Santos, Fernanda Duarte Mendes, Maria del Carmen Bisi Molina, José Geraldo Mill, Carolina Perim de Faria

**Affiliations:** 1Programa de Pós-Graduação em Nutrição e Saúde, Federal University of Espírito Santo, Vitoria 29075-910, Brazil; taiz_karla@hotmail.com (T.K.B.M.); fernanda.mendes@edu.ufes.br (F.D.M.); mdcarmen2007@gmail.com (M.d.C.B.M.); 2Programa de Pós-Graduação em Saúde Coletiva, Federal University of Espírito Santo, Vitoria 29075-910, Brazil; hully.santos@edu.ufes.br (H.C.d.S.); josegmill@gmail.com (J.G.M.)

**Keywords:** non-nutritivesweeteners, artificial sweeteners, sweetening agents, diet

## Abstract

Background: Although non-nutritive sweeteners (NNSs) were formulated primarily for sugar-restricted diets, nowadays, their consumption has become widespread among the general population. Thus, the purpose of this study was to estimate the prevalence of the regular use of NNSs and their associated factors among non-diabetic individuals from the baseline of the Longitudinal Study of Adult Health (ELSA-Brasil). Methods: In total, 9226 individuals were analysed, and the regular consumption of NNSs was defined as follows: NSSs are used at least once a day. Associations between exposure and outcomes were analysed using chi-square and Student’s t-tests. Significant variables were inserted into a binary logistic regression model to determine the adjusted association measures (significance level of 5%). Results: The prevalence of regular NNS consumption was 25.7%. Regular NNS consumption increased with age, categories of BMI, income, and schooling. The odds of regularly consuming NNSs were 1.9-times higher among women, 6.1-times higher among obese individuals, and 1.8-times higher among those with higher schooling and income. Conclusions: Specific groups seem to present a larger association relative to the use of NNS. Based on the significant overall prevalence of the regular use of NNSs, a discussion regarding country-wide policies related to their intake is needed to address recent WHO directions concerning this additive.

## 1. Introduction

Non-nutritive sweeteners (NNSs), also known as dietary sweeteners, are compounds made up of natural or artificial sweeteners, and they aim to provide a sweet taste to food by partially or completely replacing sugar [1]. Until 1988, sweeteners were considered medicines, and they were registered by the Brazilian National Division of Sanitary Surveillance of Medicines/Ministry of Health (DIMED/MS). Currently, they are defined as dietary foods, and they are registered by the National Division of Sanitary Surveillance of Food/Ministry of Health (DINAL/MS) [2].

Whilst historically consumed by diabetics, the increased availability of products containing NNSs, such as beverages, food, supplements, medicines, and hygiene products, escalates access to and consequent consumption among the general population [3]. Furthermore, despite being initially formulated for diets that advocate the restriction of simple sugars, the consumption of non-nutritive sweeteners is increasingamong the general population focusing on sugar or calorie restriction [4,5].

Although approved by regulatory agencies in many countries, including the Joint Food and Agriculture Organization (FAO)/World Health Organization (WHO) expert committee on Food Additives (JECFA), the World Health Organization has recently published a guideline discouraging the use of NNSs for the purpose of weight control or possible reductions in the risk of incident non-communicable chronic diseases (NCDs) [6,7].

Given the scenario regarding the consumption of NNSs by the general population, it is important to identify the profile of the consumers of these products in order to help researchers track individuals who are exposed to NNSs and aid public health officials in tailoring recommendations based on NNS consumption. Therefore, the present study aims to estimate the prevalence of the regular use of NNSs and their associated factors among non-diabetic individuals from the baseline of the Brazilian Longitudinal Study of Adult Health (ELSA-Brasil).

## 2. Materials and Methods

ELSA-Brasil is a multicentre cohort composed of 15,105 active or retired public servants aged 35 to 74 years, and they are from six higher education and research institutions in the northeast, south, and southeast regions of Brazil [8].

This cross-sectional study used data from the baseline of ELSA-Brasil, which was conducted between 2008 and 2010. The establishment of the cohort unfolded in two distinct phases: Stage 1, termed pre-enrolment, encompassed several pivotal components, including participant recruitment, which involved confirming individuals’ interest in participation, assessing their eligibility, and collecting essential identification data. Moreover, this phase entailed the completion of the informed consent form and the initial section of the interview. These interactions transpired either at the participants’ workplaces or at the central premises of the investigation centre (IC) [9].

Subsequently, Stage 2, known as enrolment, featured a continuation of the interview process coupled with an array of measurements and tests. Notably, all of these activities were consistently conducted at the investigation centre (IC). It is worth emphasizing that the participant recruitment strategy employed a dual approach: general strategies were implemented for standardization across multiple centres, while local strategies were tailored to accommodate the distinctive institutional contexts and the unique demographic profiles of each centre’s population [9].

The routines at the research centres followed standardized criteria, including fasting for an average of 12 h, followed by face-to-face questionnaires and other non-fasting exams. More information on measurements and clinical exams is available in the publication by Mill et al. [10].

Exclusion criteria: Individuals who did not answer the food consumption section of the study; participants who underwent bariatric surgery; those who had changed their food consumption for any reason in the six months preceding the study; participants whose energy consumption was considered implausible (<500 kcal or >6000 kcal) [11]; and individuals with laboratory diagnoses, previous diagnoses, or the use of medications for Diabetes Mellitus. All were considered ineligible for the analysis. These exclusions were implemented to uphold methodological rigour; guarantee regular exposure to NNSs and data quality; and avoid participants who might have a medical reason for using NNSs.

The exclusion criteria were based on the fact that post-bariatric surgery patients undergo numerous changes in eating habits that impact their food consumption, and they may develop food intolerances that also affect consumption [12]. These modifications may not reflect the usual food intake of these individuals.

Those who had changed their eating habits in the last 6 months would not characterize regular habitual consumption in the study population rigorously since it would not be possible to measure whether this change occurred 3 months or 15 days ago. Thus, the exclusion aimed to ensure that NNSs were being used regularly for at least six months.

Extreme energy intakes (kcal) were excluded as they were considered implausible, either due to underestimation or overestimation of intake [13]. Diabetic individuals were also excluded because disease control often involves changes in eating habits and lifestyle, frequently leading to the substitution of sugar with NNSs.

Outcome definition and independent variables:

To evaluate food consumption, a Food Frequency Questionnaire (FFQ) was used, consisting of 114 items and 14 check questions (used to confirm the frequency of certain habits, such as eating out, number of meals a day, and consumption of ultra-processed food items), aiming to assess participants’ habitual consumption in the past 12 months. The development and validation of the FFQ are described in detail in the articles by Molina et al. [14,15].

Regular consumption of non-nutritive sweeteners (NNSs) was considered when participants used at least one NNS-sweetened product (such as soda, coffee, natural fruit juice, processed fruit juice, artificial juice, or yerba mate tea) at least once a day. The consumption of sweets, cakes, and other food items was not considered since the FFQ question does not differentiate between diet and light versions, which could lead to inaccuracies in determining regular NNS consumers because Brazilian legislation allows for food items to be labelled as light if a 25% reduction in fats, salt, sugar, or others is present. In the case of sodas, although the question does not differentiate between diet and light versions, both versions contain non-nutritive sweeteners in their composition [16], so this product was considered.

Anthropometric measurements of weight and height were taken. In ELSA-Brasil, all anthropometric measurements were taken by trained research assistants according to established techniques [17]. Body Mass Index (BMI) was calculated and classified according to the cut-off points recommended by the WHO (<18.5, underweight; 18.5–24.9, normal weight; 25–29.9, overweight; >30, obesity) [18].

Sociodemographic characteristics were collected through a questionnaire and includedgender; age, divided into age groups of 35–44, 45–54, 55–64, and 65–74 years; level of education, classified as elementary, high school, and higher education, based on the highest level of education achieved; per capita family income, based on the total net income of all family members in the last three months divided by the number of dependants, categorized according to the minimum wage in force at the time of analysis—BRL 937.00; and self-reported race/skin colou0r (white, black, yellow, mixed, indigenous) according to the categories of the Brazilian Institute of Geography and Statistics [19].

Regarding lifestyle, the study variables included: self-reported alcohol and tobacco consumption and information regarding leisure-time physical activity, measured using the International Physical Activity Questionnaire (IPAQ) long version, validated for Brazil [20], classified as low, moderate, and high intensity according to the duration and intensity of the described activities.

Statistical analysis: The study variables were described using measures of central tendency (mean) and measures of dispersion (standard deviation—SD) for continuous variables and percentages for categorical variables. Student’s *t*-test and ANOVA followed by Tukey’s test were utilized to analyse differences in means, and the chi-square test was used to analyse differences in proportions. The normality of continuous variables was not tested due to the sample size, as large samples tend to follow a normal distribution [21].

Variables that reached significance (*p* < 0.05) in the bivariate analyses were included in a binary logistic regression model to determine adjusted association measures. The significance level for all tests was set at α = 0.05. Statistical analyses were performed using SPSS for Windows software, version 23.

ELSA-Brasil was approved by the National Commission of Research Ethics (CONEP) and the research ethics committees of all participating centres. Each volunteer signed an informed consent form to participate in the study.

## 3. Results

A total of 5879 participants were excluded according to pre-established exclusion criteria (Figure 1), resulting in a sample of 9226 eligible individuals.

The prevalence of the regular consumption of NNSs in the studied population was 25.7% (95% CI 24.8–26.6%), with coffee being the most frequently consumed source (20.9%) (Table 1).

Table 2 presents a description of the individuals, as well as the prevalence of non-nutritive sweetener (NNS) use according to sociodemographic and lifestyle characteristics. Women were the highest consumers of NNSs, with a prevalence of 30.5%. It is noted that 53.1% of the sample consisted of females, with the majority being classified as white (54.8%), and 39.7% of the participants were in the age group of 45–54 years, most of whom were married (67.5%).

Regarding BMI, 39.2% of the study participants were classified as overweight, the majority of whom denied a family history of diabetes (64.5%) and reported no previous diagnosis of hypertension (68.3%).

Furthermore, in Table 2, it can be highlighted that 60.7% of the respondents had a higher education level, and 35.9% had a monthly per capita family income greater than 2 minimum wages. As for lifestyle, 80.3% of the individuals studied consumed alcohol, 57.1% were non-smokers at the time of data collection, and 78.3% were classified as having a low level of leisure-time physical activity. There was a progressive increase in the prevalence of NNS use with age and BMI categories (Table 2).

In Table 2, a direct, unadjusted association between education level, per capita family income, and regular use of NNSs can be seen. Additionally, being unmarried, having a family history of diabetes, having hypertension, and consuming alcohol were favourable situations for NNS consumption. On the other hand, there was no association with tobacco use (Table 2).

An adjusted exploratory analysis of factors associated with regular NNS use in the ELSA-Brasil population is presented in Table 3. Women had nearly double the odds of using NNSs regularly compared to men, and amongst individuals with white skin colour, the odds of regular NNS use were 50% higher than amongst those with black skin colour.

Regarding age and BMI classification, in both cases, the chances of being a regular NNS user were higher as the age group and BMI category increased. The chances of usage were 1.4-times higher among those aged 65–74 years and 6.1-times higher among obese individuals compared to lean individuals.

Having a family history of diabetes, having systemic arterial hypertension, and being a consumer of alcohol increased the probability of regular NNS use by 20% among the studied population. Similarly, having a moderate or intense level of leisure-time physical activity increased the chances of NNS use by 30%.

The variables related to education level and per capita family income showed similar results. In both cases, the higher categories, namely higher education and per capita family income greater than two minimum wages, increased the chances of regular NNS use by 80% compared to the lower categories, which were elementary education and per capita family income up to one minimum wage.

## 4. Discussion

The prevalence of daily regular consumption of NNSs was 25.7% (95% CI 24.8–26.6) and was more prevalent among women, older individuals, and those with higher education. Recent studies conducted in other countries indicate different consumer profiles and higher prevalence [22,23]. A population-based cohort study in France reported a prevalence of 37.1% for NNS consumption, which was more frequent among younger individuals, those with higher BMI, and those who reported following a weight-loss diet. However, in Portugal, women, individuals with higher education, and those who are overweight and obese are more exposed to NNS consumption [23].

In Brazil, Zanini et al. [5] found a prevalence of regular use (at least 4 days a week) of artificial sweeteners of 19% (95% CI: 17.1; 20.9) in adults from Pelotas, Rio Grande do Sul. The findings of this study corroborate those found in the present research, as the prevalence of NNS consumption was higher among women and individuals with white skin colour. Additionally, a linear association between NNS use and age and education level was observed [5].

Another analysis using data from the National Survey on Access, Use, and Promotion of Rational Use of Medicines (PNAUM, 2014) in Brazil reported a prevalence of 13.4% (95% CI: 12.5–14.3), higher among females, individuals aged 60 years or older, and those in higher socio-economic categories and BMI [24].

In the United States, a cohort study with 64,850 women found a prevalence of 22.56% for consuming artificially sweetened beverages one to six times per week. When the frequency was at least once a day, the prevalence was 7.62%. The use of these products was higher among white women and showed a significant dose–response relationship with age, educational level, and family income [25].

Research using data from the Multi-Ethnic Study of Atherosclerosis (MESA), a population-based study with 6814 Caucasian, African American, Hispanic, and Chinese American adults aged 45–84 years, found that approximately 14% of participants consumed diet soda once or more a day (19.4% of white people, 8.6% of black people, 11.9% of Hispanics, and 5.4% of Chinese people) [26].

The high prevalence of regular NNS consumers among the ELSA-Brasil population, compared to data from other studies, can be partially justified by the higher contribution of NNS consumption through sweetened coffee (20.9%) [27]. Coffee consumption is widespread in the Brazilian population, and Brazil is the second largest consumer of coffee. The average coffee consumption per day in Brazil is 163 mL, and among ELSA-Brasil’s participants, it is approximately 149.45 mL per day, approaching the national average. Furthermore, 1/4 of the study population uses artificial sweeteners to sweeten their coffee [28].

The predominance of women among NNS consumers can be justified by their greater dissatisfaction with body image, as well as their greater concern for aesthetic characteristics and health aspects. These characteristics, combined with marketing strategies and the promotion of NNSs and diet/light products as health promoters and weight-loss aids, may motivate such behaviour among women [29,30].

In this way, the marketing associated with NNSs, as a component of a healthier lifestyle, causes individuals who engage in physical activity, a practice associated with a better quality of life and health, to have a higher prevalence and likelihood of consuming NNSs [29,30].

Current alcohol consumption is directly associated with regular NNS consumption. This finding may be explained by the fact that routine consumption of small amounts of alcoholic beverages has been encouraged as part of a healthy lifestyle, as it is associated with health consequences [31]. Thus, it may be part of the lifestyle practised by regular NNS users. However, this statement should be viewed with caution since the current alcohol consumption variable cannot differentiate the types and quantities of beverages consumed.

Regarding age, the direct association found in the current study and in previously mentioned research indicates that age can influence the use of NNSs [24]. This association can be explained by the relationship between the presence of chronic diseases, such as hypertension and obesity, and advancing age [32]. Since these diseases are associated with a dietary profile characterized by increased consumption of sugar, sodium, and fats, one of the recommended dietary changes for their prevention and/or treatment is a reduction in sugar consumption and the promotion of healthy eating [33].

As for a family history of DM being associated with greater odds of being a regular consumer of NNSs, there may be at least two directions for discussion: the first one is related to the family habit or food environment that could lead individuals to opt for a determined product since it is regularly consumed in one’s household; on the other hand, the second direction relies on the reverse causality of a DM diagnosis in the family causing individuals to adhere to NNSs as a strategy to avoid future disease [34].

Concerning per capita family income and education level, both variables are related and can reflect purchasing power and knowledge about health, similarly influencing the use of NNSs. It is observed that individuals with higher per capita family income and higher educational levels had a higher prevalence of regular NNS consumption in this study, as well as in the previously mentioned studies [5,6,7,8,9,10,11,12,13,14,15,16,17,18,19,20,21,22,23,24,25].

This finding may be justified by the fact that individuals with higher per capita family income and higher education seek/consume more information about food, health, and aesthetics. Although currently controversial, the idea that consuming products with NNSs is a healthy practice is precisely the focus of marketing actions for these products. Consequently, the majority of purchases of these products are still made by the upper-middle and upper classes [35].

The definition of race/colour in Brazil is still linked to social, economic, and cultural contexts. The influence of race on social inequalities varies between countries, but even in different proportions, there remains a particular dimension of social stratification that defines differences in access to goods and services. For example, “black” individuals are more likely to be poor than “white” individuals. Therefore, as white individuals tend to be in higher income classes and, as mentioned earlier, the majority of NNS purchases are made by the upper-middle and upper classes, the higher prevalence and likelihood of NNS consumption are among white individuals [36].

In the present study, despite using a semi-quantitative Food Frequency Questionnaire (FFQ) that relies on the memory of past dietary habits and well-trained interviewers, the same was validated for the study population, and training and quality control of the interviewers were carried out. Additionally, to characterize habitual/regular consumption among individuals, those who changed their dietary habits in the last six months were excluded from the study. This was meant to ensure that the analysed data reflected the habitual consumption of the study participants. Furthermore, there is no fixed definition of the regular use of NNSs, which might demand cautiousness to make comparisons, but our definition of regular NNSs aimed for greater sensitivity focusing on a minimum daily consumption of these additives.

## 5. Conclusions

Certain demographics appear to exhibit a stronger association with the utilization of non-nutritive sweeteners (NNSs). These patterns of usage may be of interest for public health monitoring in order to track trends in dietary habits as well as to advance research on the impacts of NNSs on general health. It may also be of special importance for marketing strategies, product development, and labelling regulations and guidelines.

The relatively commonplace ingestion of non-nutritive sweeteners (NSSs) within a significant segment of the population bears implications for policy and practice, not only within Brazil but also across diverse nations. In this context, where changing dietary habits and patterns significantly influence health outcomes, the proliferation of products containing NSSs, notably those described as highly processed, raises concerns about the quality of a nation’s diet and urges the need for strategies focused on transparent food labelling, consumer education, and public health initiatives focused on Food and Nutritional Security.

This approach could also serve as a model for other countries dealing with similar challenges. Through international collaboration and research, countries can develop tailored policies that consider cultural, dietary, and health attributes in order to promote diets that are neither rich in free sugars nor NNS additives.

## Figures and Tables

**Figure 1 nutrients-15-04785-f001:**
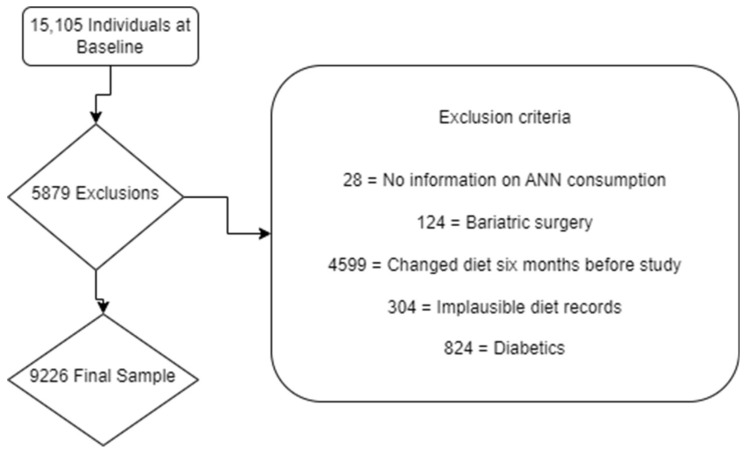
Flowchart of exclusions.

**Table 1 nutrients-15-04785-t001:** Number of participants of ELSA-Brasil ^1^ referring daily consumption of products containing non-nutritive sweeteners (2008–2010).

Product	Number	%
Soda	344	3.7
Coffee	1928	20.9
Natural Fruit Juices	348	3.8
Processed Fruit Juices	174	1.9
Artificial Juices	78	0.8
Yerba mate tea	238	2.6

^1^ ELSA-Brasil, Brazilian Longitudinal Study of Adult Health.

**Table 2 nutrients-15-04785-t002:** Description of the sample and prevalence of regular use of non-nutritive sweeteners among participants of ELSA-Brasil ^1^, according to demographic, socioeconomic, and health characteristics (2008–2010).

Characteristics	Sample		NNS ^2^ Use		*p*-Value ^3^
	N	%	Prevalence	IC95%	
**Gender**					
Male	4326	46.9	20.3	19.1–21.5	<0.001
Female	4900	53.1	30.5	29.2–31.8	
Skin Color/race					
Black	1292	14.2	18.5	16.5–20.7	<0.001
Brown	2522	27.7	22.0	20.4–23.7	
White	4996	54.8	29.4	28.2–30.7	
Yellow/Indigenous	305	3.3	25.6	21.0–30.8	
**Age group**					
35–44	2047	22.2	20.4	18.7–22.2	<0.001
45–54	3661	39.7	24.4	23.1–25.8	
55–64	2571	27.9	29.5	27.8–31.3	
65–74	947	10.3	32.0	29.1–35.0	
**Marital Status**					
Married	6229	67.5	25.0	23.9–26.1	0.017
Unmarried	2997	32.5	27.3	25.7–28.9	
BMI ^4^ classification					
Underweight	115	1.2	8.7	4.8–15.3	<0.001
Eutrophy	3817	41.4	20.1	18.9–21.4	
Overweight	3612	39.2	28.4	27.0–29.9	
Obesity	1677	18.2	34.0	31.8–36.3	
**Family history of DM** ^5^					
No	5868	64.5	24.8	23.7–25.9	0.001
Yes	3224	35.5	27.9	26.3–29.4	
**Presence of Hypertension**					
No	6295	68.3	24.6	23.6–25.7	<0.001
Yes	2926	31.7	28.1	26.5–29.7	
Education level					
Elementary school	893	9.7	15.1	12.9–17.6	<0.001
High school	2737	29.7	18.9	17.5–20.4	
College/University degree	5596	60.7	30.8	29.6–32.0	
**Monthly household income per capita**					
Up to 1 MW ^6^	2977	32.4	16.2	14.9–17.6	<0.001
From 1 to 2 MW	2915	31.7	26.3	24.7–27.9	
More than 2 MW	3300	35.9	33.8	32.2–35.5	
**Current alcohol consumption**					
No	1631	19.7	20.8	18.9–22.9	<0.001
Yes	6662	80.3	26.8	25.8–27.9	
**Tobacco consumption**					
No	5269	57.1	26.2	25.0–27.4	<0.244
Yes	3957	42.9	25.1	23.8–26.5	
**Leisure time physical activity**					
Weak	7122	78.3	24.4	23.4–25.4	<0.001
Moderate	1199	13.2	31.3	28.7–34.0	
Intense	780	8.6	30.5	27.4–33.8	

^1^ ELSA-Brasil, Estudo Longitudinal de Saúde do Adulto-Brazil. ^2^ NNS, Non-nutritive sweeteners. ^3^ Chi-square test. ^4^ BMI, Body Mass Index. ^5^ DM, Diabetes Mellitus. ^6^ MW, Minimal Wage.

**Table 3 nutrients-15-04785-t003:** Odds ratio (OR) of regular consumption of non-nutritive sweeteners according to sociodemographic and lifestyle characteristics. ELSA-Brasil ^1^, 2008–2010.

Characteristics	OR (CI 95%)		*p*-Value
**Gender**			
Male	1.0		
Female	1.9	(1.7–2.1)	<0.001
**Skin color/race**			
Black	1.0		
Brown	1.3	(1.0–1.5)	0.024
White	1.5	(1.2–1.7)	<0.001
Yellow/indigenous	1.2	(0.8–1.7)	0.379
**Age group (years)**			
35–44	1.0		
45–54	1.2	(1.0–1.4)	0.011
55–64	1.4	(1.2–1.6)	<0.001
65–74	1.4	(1.2–1.9)	0.001
**Marital status**			
Married	1.0		
Unmarried	0.9	(0.8–1.0)	0.052
**BMI ^2^ classification**			
Underweight	1.0		
Eutrophy	2.6	(1.2–5.7)	0.017
Overweight	4.7	(2.1–10.3)	<0.001
Obesity	6.1	(2.8–13.5)	<0.001
**Family history of DM ^3^**			
No	1.0		0.001
Yes	1.2	(1.1–1.3)	
**Presence of Hypertension**			
No	1.0		0.020
Yes	1.2	(1.0–1.3)	
**Educational Level**			
Elementary School	1.0		
High school	1.3	(1.0–1.6)	0.051
College/University degree	1.8	(1.4–2.4)	<0.001
**Monthly household income per capita**			
Up to 1 MW ^4^	1.0		
From 1 to 2 MW	1.5	(1.3–1.7)	<0.001
More than 2 MW	1.8	(1.4–2.4)	<0.001
**Current alcohol consumption**			
No	1.0		
Yes	1.2	(1.0–1.4)	0.026
**Leisure time physical activity**			
Weak	1.0		
Moderate	1.3	(1.1–1.6)	0.010
Intense	1.3	(1.1–1.5)	<0.001

^1^ ELSA-Brasil, Estudo Longitudinal de Saúde do Adulto. ^2^ BMI, Body Mass Index MW.^3^ DM, Diabetes Mellitus. ^4^ Minimal Wages.

## Data Availability

The data presented in this study are available on request from the corresponding author.

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
