# Peer review of "Examining the Usage Patterns of Non-Nutritive Sweeteners among Non-Diabetic Individuals: Insights from the Longitudinal Study of Adult Health (ELSA-Brasil)"

_nutrients, 2023, doi:10.3390/nu15224785_

Round 1
Reviewer 1 Report
This manuscript reports interesting results from a large-scale study. With substantial improvements it may contribute to the body of related research.
As a minor point, it is important to be consistent with the acronym used to refer to non-nutritive sweeteners.
Another minor point is that Table 2 uses both Portuguese and English. the Portuguese word is perfectly understandable to English-readers, but it would be preferable to have language uniformity.
Several more substantive points need to he addressed.
1. The authors note that "Variables that reached significance (p < 0.05) in the bivariate analyses were included in a binary logistic regression model to determine adjusted association measures." This is a fairly common practice that could lead to biased results in estimates for larger models through capitalizing on chance and not adjusting through a Bonferroni correction of false discovery rate for inflated Type I error.
2. "A total of 5,879 participants were excluded according to pre-established exclusion criteria (Figure 1), resulting in a sample of 9,226 eligible individuals." This extensive exclusion of data fairly screams for detailed checks to see if the usable sample is representative. It is not clear what efforts were made to demonstrate representativeness. Without such checks it is pretty much impossible to know whether it is appropriate to draw meaningful conclusions from the results.
3. A couple of paragraphs need to be added on implications of the findings for policy and practice in Brazil, and potentially in other countries.
4. More detailed explanations are needed of how the research was conducted and why decisions were made, to guide future researchers who could replicate or extend the findings of this study.
The English exposition generally is good, but careful editing is needed to bring the verbiage up to publishable standards.
Author Response
Dear Reviewer,
We appreciate your time and effort in reviewing our manuscript. Your valuable feedback has provided us with an opportunity to address certain concerns and enhance the quality of our work. We would like to respond by saying that:
1. We believe that the suggested tests focusing on adjusting through a Bonferroni correction of false discovery rate for inflated Type I error should be an interesting choice if we were to test a large amount of hyphoteses such as done on genomic studies. Our study focuses on around 10 variables, which does not seem to pose greater risk for inflating Type 1 error. Also, most variables that remained associated to the outcome in our adjuetd model were significant ate the 1% level.
2. Commencing with a firm affirmation, it is underscored that the sample derived from the ELSA-Brasil study is not representative of either the broader Brazilian populace or the workforce encompassed by the participating institutions. While acknowledging the concerns raised regarding the extent of exclusions, it is paramount to elucidate that the primary focus lies in ensuring consistent exposure to NSS, as opposed to sporadic utilization of the additive. To this effect, individuals who indicated alterations in dietary patterns within the preceding six months of the interview were purposefully excluded (n = 4,599). The precise nature of these dietary modifications remains uncharted; nevertheless, despite the substantial magnitude of these exclusions, their indispensability for the comprehensive integrity of the study is upheld.
3 and 4. We have completely rewritten the conclusions paragraph in order to comply with this valuable suggestion. Also, a few sentences were added in order to justify some of the choices made by our study group.
Once again, we thank you for the valuable considerations and hope our answers are considered appropriate.
Carolina P Faria
Reviewer 2 Report
The authors estimated the prevalence of regular use of non-nutritive sweeteners and the associated factors among non-diabetic individuals. Even though they get some effective conclusions, the data is too old, which can't provide references for the other researchers.
I searched other reports based on ELSA-Brasil, the recent report was published in 2018. Hence, I think the report in the manuscript is out of date.
Author Response
Dear Reviewer,
We appreciate your comments and feel like they enhance the quality of our work.
Regarding our dataset being outdated, we would like to argument that:
-
Availability of Data: At the time of conducting our study, the data we employed was the most comprehensive and suitable dataset available to us. The process of collecting, analyzing, and interpreting large-scale data can be time-consuming, and there might have been limitations in accessing more recent data within the scope of our research.
-
Historical Perspective: Although our data may not reflect the very latest trends, it does offer valuable insights into the prevalence of regular non-nutritive sweetener use and its associated factors during the time period covered. This historical perspective can still contribute to the understanding of shifts and changes in usage patterns over time.
-
Methodological Consistency: The methodology we employed remains valid, and our findings provide a foundation for future studies to build upon. Researchers interested in investigating changes over time could utilize our approach with more current datasets to compare against our results.
- Longitudinal data: ELSA-Brasil has collected data on diet only during baseline (more comprehensive questionnaire) and wave 3 (reduced questionnaire); therefore, it is important to acknowledge the our choice of using this first data set will eventually lead to longitudinal analysis using waves 3 and 4 (currently being collected).
Thank you for our understanding and kind considerations.
Prof. Dr. Carolina P Faria
Reviewer 3 Report
Thank you for this excellent manuscript, which should be of high interest to the readership of Nutrients. This reviewer has only a few minor suggestions:
line 81: "... consisting of 114 items and 14 check questions, " Please either explain what is meant by "check questions" by adding a parenthesis and example or rewrite this phrase with different terminology so that the readership can clearly understand what is meant.
lines 136 and138: The sentence beginning "table 2 presents..." is repeated twice at the beginning of this section. Please remove one of the sentences.
line 211: In this paragraph you describe the MESA study. In your previous sections you state the location of the various example studies, in France, in Brazil, etc. Please add the location of the MESA study for clarity.
Thank you again for this excellent manuscript!
Author Response
Dear Reviewer,
We appreciate your time and effort in reviewing our manuscript. Your valuable feedback has provided us with an opportunity to address certain concerns and enhance the quality of our work. We would like to respond by saying that all topics are aknowledged and corrected on the latest manuscript submitted.
Thanks again
Carolina P Faria
Reviewer 4 Report
The reviewed paper presents interesting research results; however, it cannot be recommended for publication in its current version. The following reasons account for my inability to recommend this paper:
1. Firstly, the selection of participants for the study, despite a relatively large sample size (N=9226), raises concerns for the reviewer due to the substantial exclusion rate of approximately 40%. To ensure the study's validity, it is crucial to clarify the efficiency of participant recruitment methods and whether a statistical power analysis was performed.
2. Secondly, the cross-sectional nature of the study, including food intake data as well as Non-Nutritive Sweeteners (NNS) obtained from the years 2008-2010, should not be extrapolated to the current situation. From a scientific perspective, conducting a comparative assessment with recent years would provide more insightful implications and highlight trends and differences over time.
3. Moreover, the description of products containing NNS among consumers (Table 1) only encompasses six food groups, thus offering an incomplete evaluation of their presence in the market. Other food products, such as cakes, sweet goods, desserts, chewing gum, chewing drops, and sugar confectionery, could have been available on the market during the study period. Additionally, the study overlooks table-top sweeteners, which are often used as sugar substitutes. Consequently, the lack of comprehensive data on NNS usage among adults hinders a proper analysis and interpretation of the results.
4. The authors identified a prevalence rate of 25.7% for regular NNS intakeamong consumers, deeming it significant. It remains unclear upon what basis the authors made such an interpretation. Without a thorough justification, this claim lacks substantiation.
5. Overall, the conclusions presented by the authors lack support from the study's findings. The authors should be attentive to the following risk assessment considerations when drawing conclusions.
6. I kindly request the authors to review the below comments and take them into their revission process:
NNS, especially intense sweeteners are a group of food additives for which in the approval process, an Acceptable Daily Intake (ADI) is set for most sweeteners. The ADI are based on NOAEL that are a guideline quantity that represents the amount of NNS that can be safely consumed (mg/mg/body weight/day). ADI means that the expected exposure to the NNS used in foods at the levels necessary to achieve technological effects does not represent a hazard to health risk. The intake of NNS in the quantities within the ADI does not constitute a health hazard to consumers. When the estimated intake of NNS for average consumers was below the ADI concluding does not suggest a health concern.
Moderate editing of English language required.
Author Response
Dear reviewer,
We greatly appreciate your insightful review of our manuscript. Your thoughtful feedback has been invaluable in refining our study, and we are grateful for the opportunity to respond to your concerns regarding participant selection, exclusion rates, and our methodology.
You aptly noted the relatively high exclusion rate of approximately 40% in our study, and we acknowledge the importance of clarifying the underlying reasons for this. It is essential to underscore that the ELSA-Brasil study, from which our data is derived, was not designed to provide a representative sample of the entire Brazilian general population or even specific institutional populations. Rather, the primary objective of the ELSA-Brasil study was to investigate cardiovascular diseases and diabetes, employing a detailed assessment of various risk factors.
Here is our comprehensive response to the points you raised:
-
Efficiency of Participant Recruitment Methods: While we concur that providing additional details about our participant recruitment methods would enhance the transparency of our research, it is important to note that the ELSA-Brasil study focused on recruiting individuals within specific occupational settings, including academic institutions and research centers. These settings were selected for their convenience in reaching a cohort of participants relevant to the primary objectives of the larger study. Two parapraphs have been added to clear out the proccess of recruitment.
-
Statistical Power Analysis: We appreciate your recognition of the significance of a statistical power analysis in ensuring robust results. Although our manuscript did not explicitly mention this analysis, we would like to clarify that the study's primary aim was not to achieve a representative sample of the entire population.
-
Exclusion Rates and Justification: The substantial exclusion rate was predominantly due to our stringent criteria that aimed to identify individuals engaged in regular consumption of non-nutritive sweeteners.
-
Availability of Data: At the time of conducting our study, the data we employed was the most comprehensive and suitable dataset available to us. The process of collecting, analyzing, and interpreting large-scale data can be time-consuming, and there might have been limitations in accessing more recent data within the scope of our research.
-
Historical Perspective: Although our data may not reflect the very latest trends, it does offer valuable insights into the prevalence of regular non-nutritive sweetener use and its associated factors during the time period covered. This historical perspective can still contribute to the understanding of shifts and changes in usage patterns over time.
-
Methodological Consistency: The methodology we employed remains valid, and our findings provide a foundation for future studies to build upon. Researchers interested in investigating changes over time could utilize our approach with more current datasets to compare against our results.
- Longitudinal data: ELSA-Brasil has collected data on diet only during baseline (more comprehensive questionnaire) and wave 3 (reduced questionnaire); therefore, it is important to acknowledge the our choice of using this first data set will eventually lead to longitudinal analysis using waves 3 and 4 (currently being collected).
- With respect to alternative products which may or may not incorporate Non-Nutritive Sweeteners (NNS) and could be consumed within our studied population, our methodology stipulated their exclusion from the analysis. This decision was predicated on our inherent limitation in definitively ascertaining the presence of NNS within these products. It is pertinent to note that the regulatory framework in Brazil authorizes the employment of the term "light" to denote products featuring reduced quantities of specific ingredients such as sugar, salt, or fat. Given that our dietary data was sourced from a Food Frequency Questionnaire (FFQ), the absence of label information hindered our ability to corroborate the compositional makeup of each item. This rationale underpins our deliberate exclusion of such products from our analytical scope.
-
In substantiating our premise of a noteworthy prevalence of 25.7% concerning the regular consumption of Non-Nutritive Sweeteners (NNS), several pertinent considerations come into play. Firstly, it's noteworthy that NNS were initially classified as medicinal agents, particularly within the context of Brazil. This historical perspective underscores the significance of evaluating their dietary integration. Furthermore, our belief rests on the argument that the consumption pattern elucidated in our study is not solely attributable to medical prescription but rather reflects non-prescriptive utilization without a clinical imperative. This underscores the broader societal incorporation of NNS into dietary practices. Additionally, when juxtaposed against extant literature, specifically in relation to our criteria pertaining to regular usage, our observed prevalence emerges as comparatively elevated. This variance in prevalence is particularly discernible when considering other studies. This observation underscores the potential distinctiveness of consumption trends in our studied population. In essence, our rationale for ascribing importance to the reported prevalence of regular NNS consumption is grounded in historical, contextual, and comparative considerations, all of which lend credence to the uniqueness of our findings.
- We have rewritten our conclusions in order to comply with your suggestion.
In light of your insightful comments, we will revise our manuscript to provide clearer context regarding the ELSA-Brasil study's specific goals, its emphasis on occupational cohorts, and the rationale behind the exclusion criteria.
Your diligence in reviewing our work is greatly appreciated, and your feedback will undoubtedly enhance the transparency and credibility of our study. Should you have further questions or suggestions, please feel free to reach out to us.
Thank you for your time and valuable insights.
Round 2
Reviewer 1 Report
The authors have responded to all comments adequately.
At this stage, only routine final editing appears necessary.
Author Response
Thnak you for the review.
Reviewer 2 Report
The revised version is acceptable.
Author Response
Thank you for the Review.
Reviewer 4 Report
The authors have addressed the questions raised in the 1st round of the review. They have corrected the text according to my suggestions, I have no further comments.
Minor editing of English language required.
Author Response
Thank you for the review.